# Each Cellular Compartment Has a Characteristic Protein Reactive Cysteine Ratio Determining Its Sensitivity to Oxidation

**DOI:** 10.3390/antiox12061274

**Published:** 2023-06-14

**Authors:** Ricardo Pires das Neves, Mónica Chagoyen, Antonio Martinez-Lorente, Carlos Iñiguez, Ana Calatrava, Juana Calabuig, Francisco J. Iborra

**Affiliations:** 1Center for Neuroscience and Cell Biology, CIBB—Centre for Innovative Biomedicine and Biotechnology, University of Coimbra, 3004-517 Coimbra, Portugal; 2IIIUC—Institute of Interdisciplinary Research, University of Coimbra, 3030-789 Coimbra, Portugal; 3Centro Nacional de Biotecnología, CSIC, Darwin 3, 28049 Madrid, Spain; 4Unidad de Investigación, Innovación y Docencia Médica, Hospital Universitario Vinalopó, 03293 Elx, Spain; 5Fundación para el Fomento de la Investigación Sanitaria y Biomédica de la Comunidad Valenciana (FISABIO), 46020 Valencia, Spain; 6Department of Biotecnology, University of Alicante, 03690 Alicante, Spain; 7Department of Pathology, Fundación Instituto Valenciano de Oncología, 46009 Valencia, Spain; 8IC Biomed, 03638 Salinas, Spain; 9Instituto de Biomedicina de Valencia, CSIC, Jaime Roig 11, 46010 Valencia, Spain; 10Centro de Investigación Príncipe Felipe (CIPF), Primo Yufera 3, 46012 Valencia, Spain

**Keywords:** oxidative stress, nuclear speckles, SMN, RNA polymerase, transcription, 8-hydroxy guanosine, thioredoxin, oxidized RNA

## Abstract

Signaling and detoxification of Reactive Oxygen Species (ROS) are important patho-physiologcal processes. Despite this, we lack comprehensive information on individual cells and cellular structures and functions affected by ROS, which is essential to build quantitative models of the effects of ROS. The thiol groups from cysteines (Cys) in proteins play a major role in redox defense, signaling, and protein function. In this study, we show that the proteins in each subcellular compartment contain a characteristic Cys amount. Using a fluorescent assay for -SH in thiolate form and amino groups in proteins, we show that the thiolate content correlates with ROS sensitivity and signaling properties of each compartment. The highest absolute thiolate concentration was found in the nucleolus, followed by the nucleoplasm and cytoplasm whereas protein thiolate groups per protein showed an inverse pattern. In the nucleoplasm, protein reactive thiols concentrated in SC35 speckles, SMN, and the IBODY that accumulated oxidized RNA. Our findings have important functional consequences, and explain differential sensitivity to ROS.

## 1. Introduction

ROS are chemically reactive molecules that play important roles in living organisms; for example, a moderate increase in ROS can promote cell proliferation and differentiation [1,2] whereas excessive amounts of ROS can cause oxidative damage to lipids, proteins, RNA, and DNA [3], resulting in cellular dysfunction and cell transformation. Maintaining ROS homeostasis is thus critical for normal cell growth and survival. Therefore, understanding the effects of reactive oxygen species in biological systems is critical, both in terms of the damage they cause and their role in cell signalling. Although the biological chemistry of various oxidants is becoming better understood, it is frequently unclear how this translates into cellular mechanisms where redox changes have been demonstrated.

Electrophilic molecules are electron-deficient species that act as toxins or signalling mediators in cells [4]. These molecules form irreversible covalent bonds with electron-rich nucleophiles. These bonds are irreversible under oxidative stress conditions, resulting in cell malfunction. According to Pearson’s Hard and Soft, Acids and Bases (HSAB) theory, electrophiles form covalent bonds with nucleophiles of comparable softness or hardness preferentially and more rapidly. According to this theory, which is supported by experimental data, the nucleophiles targeted by electrophiles inside the cell are the side chains on protein amino acids, with the thiolic group of Cys in the anionic form thiolate 4 being the most reactive. Because the pKa of the thiol group of Cys in proteins depends on the neighbouring residues and the solvent [5], not all Cys will be in thiolate form at physiological pH.

For the reasons stated, the thiol group of Cys is highly reactive with ROS, which is the basis for Glutathione’s (GSH) antioxidant role. Because of Cys’s high reactivity, there was a strong selective pressure favouring its exclusion from protein surfaces to prevent ROS protein inactivation [6]. As a result, accessible Cys residues may be involved in the pathophysiology of ROS. Protein thiolic groups, similar to Cys in GSH, have been proposed to play an active role in the defense against ROS [7,8].

Cys is an important aminoacid whose redox status can affect protein structure and function. Because some thiol reactions in Cys are reversible, this amino acid is well suited for physiological modifications that play a key role in redox sensing and signalling [9,10]. Because of its ability to adopt a wide range of oxidation states, covalent modification of the thiolic group of proteins is the way to convert an oxidative signal into a biological response [11]. In this way, the cellular redox balance status influences protein redox signalling [12].

The cell contains compartments with different redox potential, such as the mitochondria where most of the oxidative processes take place. Therefore, it makes sense to assume that mitochondrial proteins have undergone pressure from evolution to reduce Cys content. According to some studies, various sub-cellular compartments may each have a distinct redox balance, supporting this theory [13]. We have not been able to create a sub-cellular high-resolution map of redox balance, though, up until this point. We investigate how protein reactive thiolic groups are distributed in the cell and the effects of such distribution on the ROS sensitivity of various compartments because the cell is highly compartmentalised with distinct and characteristic protein species in each structure.

We found that the characteristic distribution of protein reactive thiolic groups in each cell and subcellular compartment correlates with the sensitivity and signalling abilities of the cell in response to ROS.

## 2. Materials and Methods

### 2.1. Bioinformatic Proteomic Analysis

Protein sequences were obtained from/by/using the UniProt human proteome set (version 2011-07) [14]. For each protein sequence the total number of cysteine and lysine residues, as well as sequence length were measured. For each protein sequence, the relative exposure of residues was estimated with NetSurfP version 1.1 [15]. Both exposed cysteine and lysine total counts per protein were calculated. Protein sequences were associated with each cellular component of interest using the corresponding Gene Ontology terms and annotations provided by the UniProtKB-GOA project (GO version 2011-07) [16]. For each cellular component, a distribution of the C_e_/K_e_ ratio per protein sequence was obtained (where C_e_ and K_e_ are, respectively the total number of exposed cysteine and exposed lysine in the protein sequence). The mean and standard deviation of cellular component distributions are reported. Differences were tested by a Student *t*-test, using MATLAB (R2010a, The Mathworks, Inc., Madrid, Spain). *p*-values and confidence intervals at a 95% significance level are reported.

Table 1 shows the number of proteins used for the bioinformatic analysis in each compartment (size means the number of proteins in the category). Moreover, this includes the average number of aminoacids contained per protein in each category.

### 2.2. Cell Culture and Treatments

Hela cells (HeLa, from the American Tissue Culture Collection (ATCC^®^ CCL-2™), were cultured in DMEM supplemented with 10% fetal calf serum. Cells were fixed with 4% paraformaldehyde in 250 mM Hepes or PBS. Alternatively, cells were fixed with acetone at −20 °C. Paraformaldehyde fixed cells were permeabilised with Triton X100 (0.5% for 10 min at room temperature).

For anti and pro-oxidant treatments, cells were incubated in DMEM for the time and concentration described in the legend of figures with diamide or *N*-acetyl cysteine, both from (Merck Life Science S.L.U. Madrid, Spain). These reagents were prepared immediately before use.

### 2.3. Reactive Thiol Groups and Protein Staining

Reactive thiol groups were detected by incubation of fixed cells and permeabilized cells with Alexa-488 Maleimde (Mal-488) from (Thermo Fisher Scientific, Waltham, MA USA), 100 nM in PBS for 10 min at room temperature. Proteins were stained with the amine-reactive compound Alexa Fluor 647 carboxylic acid, succinimidyl ester (CCE 647) from (Thermo Fisher Scientific), 2 μM in PBS for 10 min at room temperature.

Controls using iodoacetamide were performed using 10 mM iodoacetamide in bicarbonate buffer pH 8 for 30 min.

### 2.4. Apoptosis Induction by TRAIL

Hela cells were exposed to TRAIL (30 ng/mL) for 1 h then fixed and stained with Mal-488 and CCE 647.

### 2.5. Transcription and Immunofluorescence

For “in vivo” transcription, cells were incubated in 2.5 mM BrU (Sigma) for 30 min. For RNA pol I activity, cells were incubated for 1 h with 2 mg/mL of alpha amanitin. Incubation for 1 h with 100 µM DRB or 1 µg/mL actinomycin D prior to BrU completely abolished BrU incorporation.

For “in vitro” transcription we used the technique previously described [17,18].

For the detection of primary transcripts, we used mouse anti-IdU/BrdU (5 mg/mL; Caltag Laboratories, Burlingame, CA, USA). Secondary antibodies were donkey anti-mouse IgG tagged with Cy3 (1/200 dilution; Jackson ImmunoResearch Europe Ltd., Ely, UK). DNA was stained with 200 nM TO-PRO-3 (Thermo Fisher Scientific) for 5 min, then slides were mounted in Vectashield (Vector Laboratories, Inc. Newark, NJ, USA).

### 2.6. Immunostaining and Antibodies

Fixed cells were immunostained as described in [17,18]. The antibodies used in this study came from different suppliers: Sigma (anti-C35); Santa Cruz Biotechnology (Dallas, TX, USA) (anti-PML, anti-SMN, anti-Coilin, anti-UBF, anti-B23); Bio-rad (Hercules, CA, USA) (anti-8HG and anti-thioredoxin); ViroGen (Watertown, MA, USA) (anti-GSH); abcam (Cambridge, UK) (anti-YB1,Nrf2 and P-Nrf2 (Phospho S40)).

As GSH is not retained by paraformaldehyde fixation, the immunodetection of GSH must be attributed to GSH that has reacted with proteins, therefore, GSH staining is evidence of protein glutathionylation.

The 8HG specifically stains oxidized RNA as demonstrates the abolition of the staining after digestion with 10 μg/mL RNAse A 1 h.

### 2.7. Protein Carbonylation Analysis

Fixed cells were incubated with 20 μM fluorescein-5-thiosemicarbazide (FTZ; Thermo Fisher Scientific) in 0.1 M 2-morpholinoethane sulfonic acid (MES)-Na buffer (pH 5.5) for 1 h and washed three times with PBS. Then, samples were mounted and visualised.

### 2.8. siRNA

Thioredoxin was knocked down using three validated siRNAs directed to thioredoxin from the *Silencer*^®^ Select Pre-Designed & Validated siRNA collection (#4390824) from (Thermo Fisher Scientific). The knock-down level was assessed by quantification of anti-thioredoxin immunostaining in individual cells. The siRNAs were transfected following manufactures’ guidelines (Quiagen, Madrid, Spain) and cells were assayed 48 h after transfection.

### 2.9. Microscopy and Image Analysis

Cells were observed with a Zeiss inverted microscope fitted with a confocal system (BioRad 2000). For Total probe content we used wide confocal cytometry as described in [18]. For Mal-488/CCE 647 ratio imaging, single confocal sections were used for each channel and the ratio image was calculated in Metamorph (ver 7.0) (Molecular Devices, MDS Analytical Technologies, San Jose, CA, USA) and data exported to Excel (Microsoft, Madrid, Spain) for analysis. DNA was stained with DAPI. Displayed images were processed (contrast stretched) and mounted in Adobe Photoshop CS4 (Madrid, Spain).

## 3. Results

### 3.1. Different Cellular Compartments Have Distinct Protein Cys Content

Because Cys is a redox-sensitive aminoacid, it is reasonable to believe that the Cys content of proteins in different compartments may reflect their redox activity and/or sensitivity.

In fact, the cell contains compartments with varying redox potentials, such as the mi-tochondria, where the majority of the oxidative processes occur. It is reasonable to conclude that mitochondrial proteins have been subjected to evolutionary pressure to reduce Cys content. To see if the Cys content of mitochondrial proteins differed from proteins from other cellular compartments. We performed a bioinformatics analysis in which we sorted proteins by subcellular location and then quantified the Cys content, as described in the Section 2. The findings indicate that the proteins in each compartment contain a specific amount of Cys. As expected, the Cys content in the global pro-teome was underrepresented (3.4 versus the expected 5%). Extracellular compartment and cytoskeleton proteins contain more Cys than the average protein in the proteome (3.4, 3.1, and 2.5%, respectively), while nuclear speckles, mitochondria, and nucleoli contain less (1.7, 1.9, 1.9, and 2.5%, respectively) (Figure 1 and Table 1). In summary, some compartments have twice the amount of Cys per protein as others. The different Cys content in proteins in each sub-cellular compartment may reflect redox heterogeneity in each structure and, as a result, influences the function and/or activities in these structures. The investigation of such a possibility necessitates the development of a method for studying Cys reactivity in various subcellular compartments.

To localise and quantify the content of reactive Cys in proteins, we used a fluorescent method. As described in [19], the method employs the reaction of Maleimide-Alexa 488 (Mal-488) with accessible and reactive-SH groups in proteins. According to Pearson’s HSAB theory, maleimides are electrophilic molecules with one of the highest reactivity with nucleophilic thiolates. Because of this property, maleimides are useful reagents for studying the reactivity of thiolic groups in Cys.

Nonetheless, non-specific reactivity is a concern with all Maleimide dyes, which may react with amino groups in proteins when used at high concentrations or for long periods of incubation [20]. We investigated whether a shorter reaction time would reduce side reactions while still allowing complete thiol labelling. To investigate the possible contribution of amino groups in proteins to the Maleimide-Alexa 488 (Mal-488) reaction, we used carboxylic acid succinimidyl ester to block free amino groups in proteins before maleimide-dye staining, which had no effect on the intensity of the Mal-488 staining (Figure 2a).

We concluded from this control that Mal-488 does not react with amino groups under our experimental conditions. We also performed several controls to ensure that the reaction was specific for reactive thiolic groups. To investigate the potential contribution of other chemical species to the staining, we blocked protein reactive -SH groups prior to the Mal-488 staining and compared the results to untreated cells. To account for the protein variability among individual cells, we introduce in our reaction another dye specific for amino groups (which is proportional to the amount of proteins) (Alexa Fluor 647 carboxylic acid succinimidyl ester (CCE 647)); as a result, we measure the Mal-488/CCE 647 ratio.

Protein -SH group blocking was done by incubation of fixed cells with the -SH alkylating molecule iodoacetamide (10 mM). The titration of Mal-488 showed that 160 μM was enough for the detection of 90% of reactive Cys groups (Figure 2b). The analysis of treated samples showed a dependency on the unspecific staining on Mal-488 concentration, for example at 80 μM (80% Cys saturation), the non-specific staining was 4% and at 160 μM it was 6% of the signal (Figure 2c). This data shows that the method has a good specificity for Cys detection. To further study whether the technique is specific for -SH groups in proteins we use different approaches.

One possibility is that GSH contributes to Mal-488 staining; however, immunostaining with specific antibodies against GSH produced a very faint, almost negative staining (Figure 2d). This is not surprising given that formaldehyde retains small peptides poorly, if at all, leading to the extraction of non-covalently attached GSH molecules.

The perturbation of the redox status of Cys in proteins provided additional evidence for the specificity of Mal-488 staining of protein reactive thiolic groups. Thioredoxin (Trx) is a protein that maintains the reduced status of Cys in proteins by reversing disulfide bridges formed by intracellular oxidative processes [21]. Then, the experimental down-regulation of Trx by siRNA must result in an increase of disulfide bridges and a decrease of free protein thiol groups; consistently, knocking down Trx by 80% resulted in a strong reduction of Mal-488 staining (Figure 2e).

Finally, we performed another control in which we decreased the concentration of cellular thiols using diamide (for chemistry check reference [22]), which reduced Mal-488 staining (Figure 2f,g). As formaldehyde reacts with the thiol groups of proteins, we studied whether the fixation we used in this study altered the staining distribution. For this purpose, we stained unfixed Hela cells, and obtained a thiol group distribution pattern equivalent to that obtained using fixed cells (Figure 2h).

This is not surprising given that when carbonyl compounds, such as formaldehyde, react with thiol groups, hemithioacetals and thioacetals are formed. These compounds are unstable and only stabilise in the presence of excess thiol groups and water-removing systems. These circumstances are missing in our experimental setup. As a result, we believe MAL-488 destabilises hemithioacetal groups, reversing the transformation and reacting with SH groups.

To summarise, we are very confident that the Mal-488 staining reacted specifically with the reactive protein thiolic groups in our experimental setup.

### 3.2. The Intensity of Maleimide-Alexa Dye Staining Reflects the Amount of Reactive Thiols in Proteins

Not every Cys residue in a protein is reactive for a variety of reasons: some are not accessible to the probe, are involved in disulfide bonds or chelating metals, or have already been modified; additionally, even when exposed, the reactivity can be hampered by a basic pKa for the thiolic group. According to a mass spectrometric study, only 12% of the Cys in the human proteome are reactive [23]. This figure is nearly half of the estimated number of exposed cysteines in the human proteome (22.5%), implying that a significant portion of cysteines are non-reactive under normal physiological conditions. To investigate the nature of these thiolic groups’ non-reactivity, we incubated cells with the antioxidant *N*-acetyl cysteine (NAC), which resulted in an increase in Mal-488 staining (Figure 3a), implying that a fraction of the exposed Cys in proteins are non-reactive due to being modified or participating in disulfide bridges.

The limits of global protein Cys modification that can be translated into a biological response are difficult to establish. Nonetheless, we can attempt to define some rough boundaries, such as a lower limit dictated by cells undergoing apoptosis, a phenomenon associated with oxidative stress [24]. When we induced apoptosis in cells by exposing them to TNF-related apoptosis-inducing ligand (TRAIL), the signal for Mal-488 decreased, as demonstrated by the decrease in the Mal-488/CCE-647 ratio (at least 60% reduction over the control) (Figure 3b). On the other hand, incubating cells with NAC increases the length of the cell cycle in a concentration-dependent manner (Figure 3c).

Cys thiol group is essential for redox sensing and signalling [9,10]. Because of its ability to adopt a wide range of oxidation states, covalent modification of the thiolic group of proteins is the method of converting an oxidative signal into a biological response [11]. Thus, we investigated whether the variability in the Mal-488/CCE 647 ratio observed in a healthy cell population reflects differences in ROS signalling.

The ratio of Mal-488/CCE 647 staining in our cells varied by 10% (Figure 4a). Is such minor variation significant enough to influence ROS signalling? To investigate this possibility, we looked at how the Mal-488/CCE 647 ratio relates to ROS signalling. ROS influence gene expression by phosphorylating and transporting Nrf2 to the cell nucleus, where P-Nrf2 activates ROS-responsive genes [25].

As expected, diamide exposure of Hela cells for 1 h resulted in a significant increase in the P-Nrf2 signal (Figure 4b). Then, to see if the Mal-488/CEE 647 ratio is related to redox signalling in cells growing under physiological conditions, we stained Hela cells with Mal-488, CCE 647, and P-Nrf2. The ratio of Mal-488/CCE 647 versus nuclear P-Nrf2 in individual cells revealed that cells with a low Mal-488/CCE 647 ratio had high nuclear P-Nrf2 staining (Figure 4b,c). The plot of Mal-488/CCE 647 versus P-Nrf2 (Figure 4b,c) revealed an exponential behaviour, which can be attributed to the Nrf2’s fine sensitivity to global Cys oxidation. This finding suggests that the Mal-488/CCE 647 ratio is a good indicator of cellular redox balance and that cells in the culture are actively using redox signalling.

We used the Mal-488/CCE 647 ratio to learn about the structure and function of subcellular structures. We first investigated the subcellular distribution of Mal-488 staining, which revealed a heterogeneous pattern. The nucleolus had the highest Mal-488 intensity, followed by the nucleoplasm and cytoplasm (Figure 5a). To see if this was due to the different protein concentrations in these compartments, we examined the content of reactive thiolic groups per protein in each one. The analysis of the Mal-488/CCE 647 ratio revealed that the nucleolus, despite having the highest content of reactive thiol groups, has the lowest concentration per protein (Figure 5b). However, the concentration of reactive thiol groups in a compartment has an inverse relationship. This means that when the concentration of reactive thiol groups is high, the number of reactive thiol groups per protein is low (Figure 5b). Based on these findings, we concluded that the heterogeneity in Mal-488 staining is caused by the accumulation of reactive thiol groups in a compartment, which does not always correspond to protein concentration in that organelle.

These findings are consistent with our bioinformatics analysis (Figure 1). We did, however, investigate whether there is a quantitative relationship between experimental and theoretical data. To make a fair comparison between the expected and experimental distribution of reactive thiol groups per protein (we assume that exposed Cys are reactive) (Ce), we plotted the Mal-488/CCE 647 ratio versus the reactive Cys/Lys theoretically expected in each compartment. We use Lys because it is the aminoacid with a primary amine in its side chain that CCE 647 is reacting with (Ke). The plot revealed a direct relationship (Figure 5c), which was not found when we compared the Mal-488/CCE 647 ratio to the average Cys content of the proteins in these compartments (Figure 5d). Surprisingly, the number of exposed Cys grows exponentially with the number of aminoacids in a protein (Figure 5e).

### 3.3. ROS Sensitivity Is Related to the Distribution of Reactive Thiol Groups in a Compartment

The asymmetric distribution of reactive thiol groups per protein in different compartments suggests that each has a different sensitivity to ROS and/or ROS-generating capabilities. One possibility is that the high concentration of reactive thiol groups per protein protects against nonspecific damage by buffering ROS or aldehydes produced by normal metabolism or lipid peroxidation. If this is the case, we can expect redox-sensitive functions to be restricted to subcellular structures rich in reactive thiol groups. To investigate this possibility, we used diamide to change the reactive thiol group balance. This perturbation, as expected, reduced the Mal-488 signal in all compartments (Figure 6a). This decrease, however, was not uniform and was proportional to the initial reactive thiol group content of each compartment (Figure 6a,b). This data shows that each compartment has a different intrinsic thiol reactivity, which could be related to the compartment’s ROS exposure and sensitivity to ROS. If our interpretation is correct, the nucleolus must produce more ROS than the nucleoplasm. Consistent with this interpretation, the nucleolus accumulates more oxidised (Carbonylated) proteins than the nucleoplasm, with values comparable to those found in mitochondria (Figure 6c). Furthermore, oxidised proteins build up in other compartments such as speckles, mitochondria, and cytoplasm (Figure 6c).

The second implication of our interpretation is that the differential distribution of reactive thiols must reflect variation in these compartments’ ROS activities. Therefore, we investigated how diamide affected activities in two compartments with different thiolate contents, nucleolus, and nucleoplasm. We concentrated on one of the most important nucleolar activities (ribosomal RNA production) at the nucleolus and on RNA pol II transcription in the nucleoplasm. We chose these activities because Mal-488 preferentially associates with UBF foci, where RNA pol I transcription occurs, and transcription foci, where RNA pol II active is located (Figure 6d). Diamide exposure caused a progressive decrease in transcription in both nucleoli and nucleoplasm (Figure 6e and Appendix A). In the case of RNA pol I, we performed “run on” assays using BrUTP after diamide incubation to determine whether this was a direct effect on RNA pol I activity or a secondary ribosomal RNA maturation defect (Figure 6f). To rule out a defect in the loading of RNA pol I onto ribosomal genes, as proposed [26], we performed “run on” transcription assays in which the reaction cocktail contained *N*-ethyl maleimide (NEM) to modify only the reactive thiol groups in proteins. Under these conditions, any differences in BrUTP incorporation must be due to effects on transcription elongation. DTT treatment increased RNA pol I and II elongation activities by increasing the reduction of -SH groups in proteins. -SH group alkylation with NEM reduced BrUTP incorporation in both compartments (Appendix A). These results indicate that transcription by RNA pol I and II is extremely sensitive to changes in the redox status of Cys and is most likely sensitive to ROS. Diamide had a greater impact on RNA pol I activity than on RNA pol II activity, which is consistent with our interpretation of the differential distribution of reactive thiols, in which the higher the concentration of reactive thiols in the compartment, the greater the sensitivity to redox stress.

Furthermore, diamide treatment stimulated actin polymerization, resulting in a compartment with a low total reactive thiol group content (Figure 6a,b). Some studies have shown that ROS plays an important role in maintaining a dynamic F-actin cytoskeleton, controlling neurite outgrowth, and cell signalling, which is consistent with our findings [27].

Glutathionylation of Cys has emerged as an important posttranslational modification in response to ROS, which can be used for cell signalling or protection [28,29]. We investigated the distribution of glutathionylated proteins after diamide treatment, which revealed compartmentalization. This modification occurred in proportion to the number of -SH reactive groups per protein (Figure 6g), rather than in a proportion to the total concentration of thiolate in the compartment. As a result, our findings suggest that the cytoplasm is the compartment where the majority of ROS signalling occurs, which is consistent with other studies [27].

Other nuclear compartments that are abundant in reactive thiols include SC35 nuclear speckles and Gems (which are positive for SMN; see Appendix A). Gems are connected to a new body we called IBODY. The IBODY lacked coilin, PML, SMN, PTF, or OCT1 markers while being high in reactive thiols and concentrated oxidised RNA (Figure 7a). This IBODY is transcriptionally inactive because it did not associate with nascent RNA or RNA pol II because colocalization was not seen, and its number ranges from zero to four.

Since we lack evidence of any particular activities occurring in the compartment where the SC35 nuclear speckles are located, it is more challenging to determine which activities there are that are impacted. The accumulation of oxidised RNA, however, was preferentially found at speckles under oxidative stress (Figure 7b,c).

Given that RNA is highly reactive to ROS (Appendix A), we investigated whether reactive -SH groups in proteins shielded RNA from oxidation in any way. The amount of 8-hydroxy-Guanosine (8HG) staining increased by 2 and 13 times, respectively, when we reduced the levels of protein reactive thiols by knocking down Trx by siRNA or thiols in both proteins and GSH by diamide (Figure 7d). Mal-488 staining was reduced by 80% of controls when Trx siRNA was used as a control. These findings imply that GSH is essential in preventing RNA oxidation.

## 4. Discussion

We demonstrate in this study that each subcellular compartment has a distinctive Cys content. This discovery inspired us to create a novel experimental method to investigate sub-cellular redox compartmentalization and redox balance at the single-cell level. The study’s experimental methodology enables the analysis of diverse cell populations.

This method outperforms existing approaches for the investigation of sub-cellular redox compartmentalization and redox balance at the single-cell level. Examples of localization artefacts [30,31] include the sub-cellular analysis of GSH distribution. Furthermore, the ratio of reduced to oxidised glutathione (GSH/GSSG) determines the intracellular redox balance, so simply visualising GSH is insufficient to provide a complete picture of the redox balance status of a given cell or sub-cellular compartment. This, along with the finding that reactive -SH groups in proteins account for roughly 70% of the total reactive -SH groups available in the cell according to a thorough analysis of the total cellular thiol pools [7].

Finding a single-cell reporter for redox status was our goal. Protein redox status and signalling are directly affected by the cellular redox balance [12]. The analysis will be incomplete in terms of redox balance if reactive -SH groups in proteins are simply visualised using Mal-dye staining rather than also considering GSSG species. By simultaneously staining the protein amino groups, we were able to overcome this problem. In this way, we can obtain a parameter reflecting changes in relative Cys reactivity simply by normalising Mal-dye staining by the protein content.

By examining the connection between this parameter and Nrf2 signalling, we validate the use of the Mal-dye/CCE-dye ratio as a reporter for the cellular redox balance.

Through Nrf2, which is phosphorylated and transported to the cell nucleus where P-Nrf2 activates ROS-responsive genes [25], ROS have an impact on gene expression. Our findings demonstrated that a small change in the Mal-dye/CCE-dye ratio had a significant impact on Nrf2 signalling (Figure 4c), behaving as was anticipated for a change in the GSH/GSSG ratio. This suggests that the Mal-dye/CCE-dye ratio is an indicator of the redox status of cells and subcellular structures. The use of redox-sensitive fluorescent proteins [32] has been another method for examining the status of redox balance in single cells and sub-cellular structures. Due to the introduced reactive disulfide bond’s thermodynamic stability, such a strategy is not always appropriate. This means that these sensors are best suited for studying reduced compartments, such as the cytosol and mitochondria, but are only effective in a small range of redox conditions [32]. Targeting the fluorescent reporter to the target compartment is also necessary for the success of these methods. This is a problem because we do not have enough information about the signals that are targeted to each sub-cellular compartment.

Furthermore, measuring the redox balance of multiple sub-cellular compartments in a single cell at once is not possible because the accurate determination of the redox balance depends on protein expression levels [33]. Our method makes it possible to build detailed hierarchical maps of the redox balance in various compartments, which could be helpful for understanding basic biology and for creating pathophysiologic models of diseases. It also enables the quantitative study of redox compartmentalization.

By using an imaging technique, we can also learn more about how the probes are distributed throughout various cellular compartments. In this study, we found that reactive -SH groups were distributed in proteins in a non-homogeneous manner (Figure 1 and Figure 5). The variations in reactive Cys concentrations in the cell nucleus were a startling discovery. This is intriguing because exposed Cys is the least conserved amino acid type and Cys is the most heavily selected amino acid, likely due to its high reactivity [34].

This strongly implies that the presence of reactive Cys has functional consequences. In accordance with this interpretation, the activities confined to the compartment showed sensitivity proportional to reactive Cys content (Figure 6). Our findings point to reactive Cys being concentrated to provide a reducing environment. The presence of reactive Cys compartmentalization in the nucleus suggests that this compartment is vulnerable to ROS, as some studies suggest [35]. Furthermore, each structure within the cell nucleus has a distinct ROS sensitivity.

ROS can induce protein carbonyl derivatives via direct metal-catalyzed oxidation attacks on carbonylatable amino acid side chains. Hydroxyl radicals, formed by the Fenton reaction of H_2_O_2_ with iron, are the only ROS capable of oxidising a car-bonylatable site [36]. Carbonylated proteins accumulate at the nucleolus, implying that the nucleolus must contain iron, and ribosomal RNA is indeed bound to redox-active iron [37]. The other component of this reaction, H_2_O_2_, can diffuse from the cytoplasm or be generated in the nucleoplasm by the flavin-dependent histone de-metylases LSD1 and LSD2. The reaction of these enzymes produces two products: H_2_O_2_ and formaldehyde [38,39]. Therefore, the nucleolus example is consistent with the idea that Cys is concentrated to create a reducing environment to safeguard its activities from the oxidative damage of ROS. According to this interpretation, it has been proposed that the nucleolus serves as a cellular stress sensor [40,41,42]. Perhaps because reduced Cys are crucial for p53 function, p53 accumulates at the nucleolus under stressful circumstances [43]. Gem bodies provide additional evidence in favour of reactive Cys’ protective function. SMN protein, which causes spinal muscular atrophy, a motor neuron degenerative disease, is concentrated in Gem Bodies [44]. The biogenesis of small nuclear ribonucleoproteins, the main building blocks of the spliceosome, requires the oligomeric protein SMN. The SMN complex is indeed a redox-sensitive complex and a ROS target [45], as predicted by our hypothesis. Nuclear detection of oxidised RNA that accumulates at SC35 speckles provides another illustration of nuclear compartmentalization. The molecular process causing the buildup of oxidised RNA in the speckles is unknown to us. However, 8HG-containing RNA can be bound “in vitro” [46] by hnRNP D, hnRNP C, and SF3B49. It’s possible that this interaction aids in the oxidised RNA’s targeting of the speckles. The justification is that hnRNP C cannot leave the nucleus [47] and localise in speckles [48], and that all of these proteins can interact with one another to form a complex. Nuclear oxidised RNA is kept in the cell nucleus in this manner, eventually making its way to the speckles. There may be some sort of repair system for oxidised RNA, but we are unaware of it.

What is apparent is that oxidised RNA has a lower stability than normal RNA, which raises the possibility of a quality-control mechanism [49]. This evidence, along with the accumulation of oxidised RNA at speckles and its instability, makes the idea that speckles serve as sites for RNA quality control very appealing. The concentration of TRAP150, a protein involved in RNA degradation, at the SC35 nuclear speckles [50], which likely scan the RNA, is consistent with this interpretation.

This study sheds light on the critical issue of the nature of the antioxidant systems in the cell, with a focus on the cell nucleus. It has been proposed that GSH concentrates in the nucleus to protect its constituents [51,52]. Our data, however, call this assertion into question. First, even if GSH concentrates in the cell nucleus, it is difficult to imagine a mechanism for this differential partitioning. GSH is much smaller than the nuclear pore exclusion size, which is around 40 kDa [53]; thus, GSH must freely diffuse in and out of the cell nucleus. Second, even if we accept that GSH concentrates in the cell nucleus, this does not imply that this compartment has a lower oxidative status than the cytoplasm, especially since we do not know anything about its GSSG counterpart.

Third, because the cell nucleus contains more reactive thiolic groups in proteins than the cytoplasm, we would expect a higher level of glutathionylation in the nucleus than in the cytoplasm, which is not the case (Figure 6g). All of these arguments make it very unlikely that GSH serves as the primary defense system against ROS in the cell nucleus. GSH appears to play a role in restoring the reduced status of Trx, detoxification, mixed GSH reactions, and the prevention of DNA and RNA oxidation.

This study also suggests that the cell has evolved two strategies for dealing with ROS at the sub-cellular compartmental level: one for signalling (high Cys per protein but low total thiolate content) and another for ROS protection (high thiol content but low number of Cys per protein). Both strategies make intuitive sense. When signal transmission is critical, a high concentration of receptors in the antenna (9 Cys in the case of the cytoskeleton) appears to be the logical way to go. When protecting against the harmful effects of ROS is important, however, an increase in protein content ensures a high concentration of thiolates in the compartment (3 Cys per protein on average) (Figure 5e). In these conditions, if some proteins are inactivated by ROS, it leaves more proteins in the structure that can function normally.

## 5. Conclusions

In summary, we show that proteins with different Cys content accumulate in different cellular compartments, and we also present a novel method for studying redox cell biology. We developed a new parameter (Mal-dye/CCE-dye) to investigate cellular redox balance at the single-cell and sub-cellular levels. This technique enabled us to identify a new nuclear body and possibly a new function for nuclear speckles in RNA proofreading. One advantage of this method over others is that it can predict the range of susceptibility to ROS damage for each cell compartment. We think that research of this kind contributes to the development of systems-level models of ROS cellular action and more thorough predictions of ROS targets.

## Figures and Tables

**Figure 1 antioxidants-12-01274-f001:**
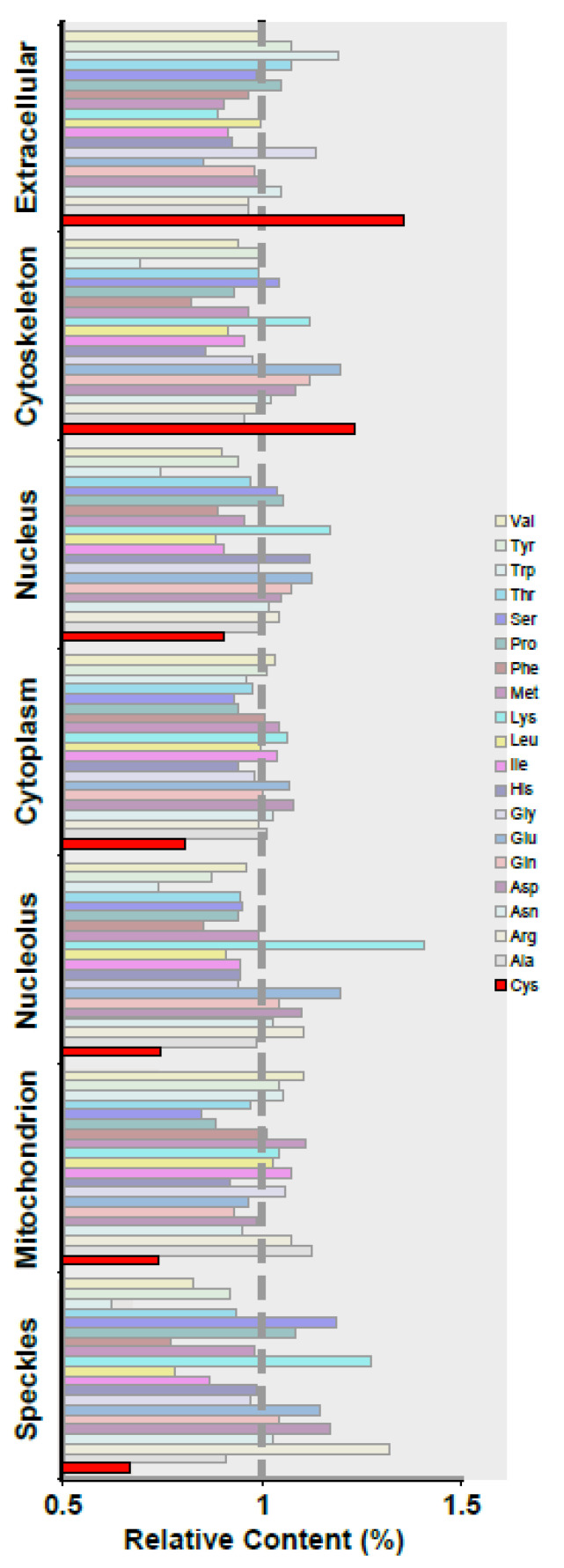
Cys distribution analysis in proteins from various compartments. Bioinformatic analysis of the aminoacid content of proteins from various cellular compartments, as standardised by proteome composition. As a result, a value greater than 1 indicates that the aminoacid is overrepresented in the compartment. The letter Cys is highlighted in red. Cys is overrepresented in the extracellular compartment and the cytoskeleton, but is underrepresented in mitochondria and SC35 nuclear speckles. 2.2. Maleimide-Alexa Dye staining is specific for reactive thiolic groups in proteins.

**Figure 2 antioxidants-12-01274-f002:**
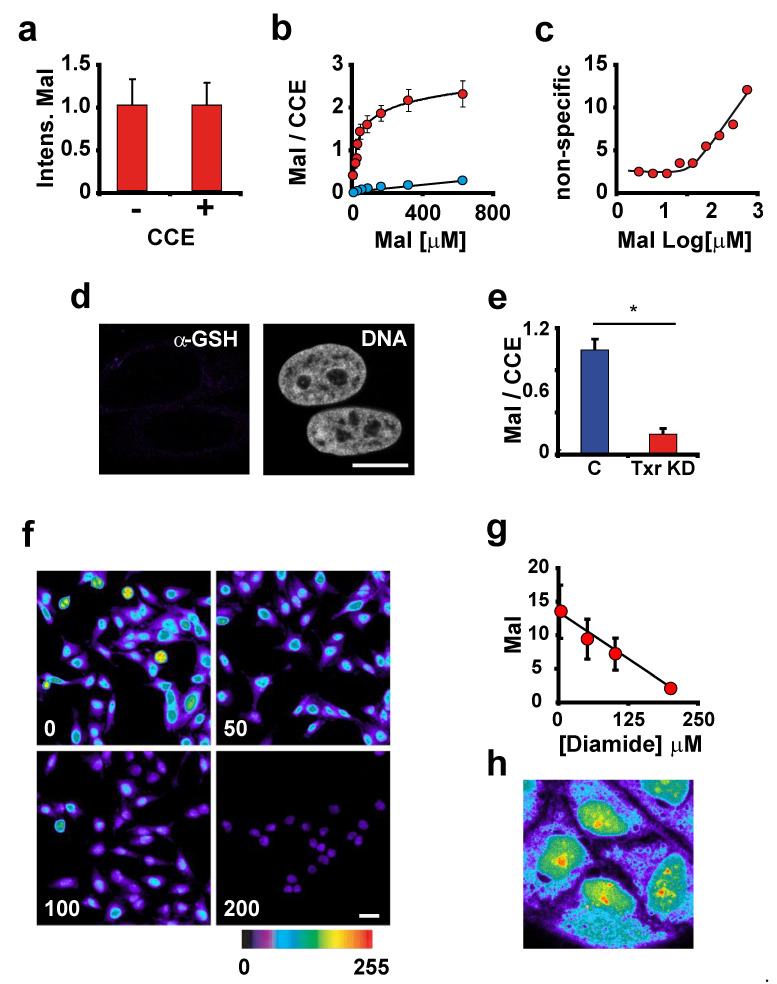
A new method to study protein thiols compartmentalization. (**a**) Blocking of amino groups. To see if Mal-488 reacts with amino groups in proteins, we blocked amino groups in fixed and permeabilized cells with CEE before staining with Mal-488 (100 nM). The intensity of the Mal-488 staining in cells incubated with CCE (+) was identical to that in cells not incubated with CCE (−). (**b**) Mal-488 titration and specificity. We titrate the concentration of Mal-488 and measure the intensity as a function of [Mal-488] to determine the degree of reactive Cys saturation. To avoid changes in Mal-488 intensity caused by changes in protein content, we plotted the Mal-488/CCE-647 ratio. The red dots represent control cells, while the blue dots represent cells treated with 10 mM iodoacetamide 30 min. (**c**) In order to calculate the specificity of Mal-488 staining we plotted for all the concentrations tested the ratio of iodoacetamide cells versus control. (**d**) Cells stained with GSH-specific antibodies revealed no signal, indicating that free GSH is extracted after fixation. Bar 10 μm. (**e**) To investigate the specificity of Mal-488 staining further, Thioredoxin was knocked down to 20% of its normal value for 48 h using siRNA. Trx reduction resulted in a significant reduction in Mal-488 staining. This decrease was not due to a possible global decrease in cellular protein content, as evidenced by the significant decrease in the ratio of Mal-488 and CCE-647 in co-stained cells with these two markers (* *p* > 0.005). (**f**) Hela cells were exposed for 2 h to various concentrations of diamide (0, 50, 100, and 200 M). As the concentration of diamide increases, the signal of Mal-488 decreases. Bar 10 μm. (**g**) Quantification of the Mal-488 signal throughout the cell. GSH depletion resulted in a direct decrease in the amount of reduced Cys in the cell. (**h**) Distribution of Mal-488 staining in unfixed cells permeabilised with 100 μg/mL saponin.

**Figure 3 antioxidants-12-01274-f003:**
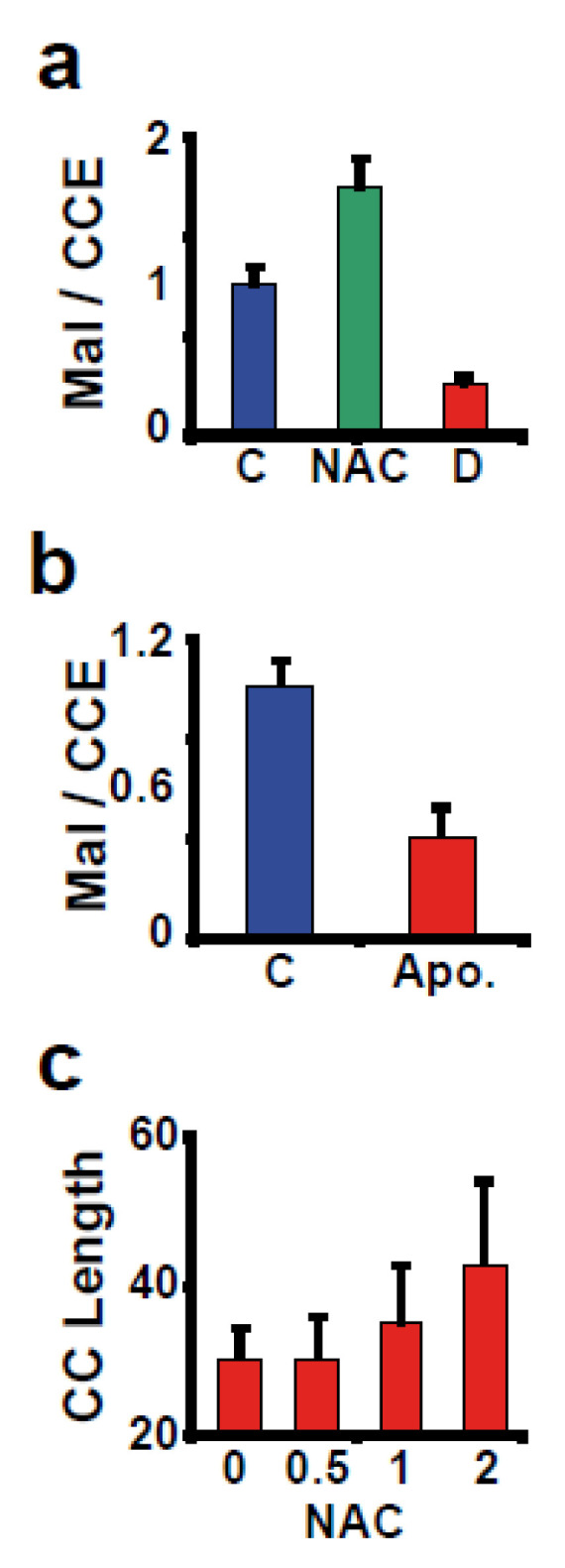
Not all of the exposed cysteines are reactive. (**a**) The increase in the ratio for Mal-488 and CCE-647 after 1 h of 1 mM NAC incubation demonstrated that cellular proteins are not completely reduced. As expected, treatment with 100 μM diamide (D) reduced the ratio of these two markers. (**b**) Apoptotic cells (Apo.) had a lower content of reactive Cys in proteins than controls. (**c**) The cell’s reduction status affects the length of the cell cycle (hours). Cells were incubated continuously in the presence of NAC (concentration in mM). The experiment lasted four days.

**Figure 4 antioxidants-12-01274-f004:**
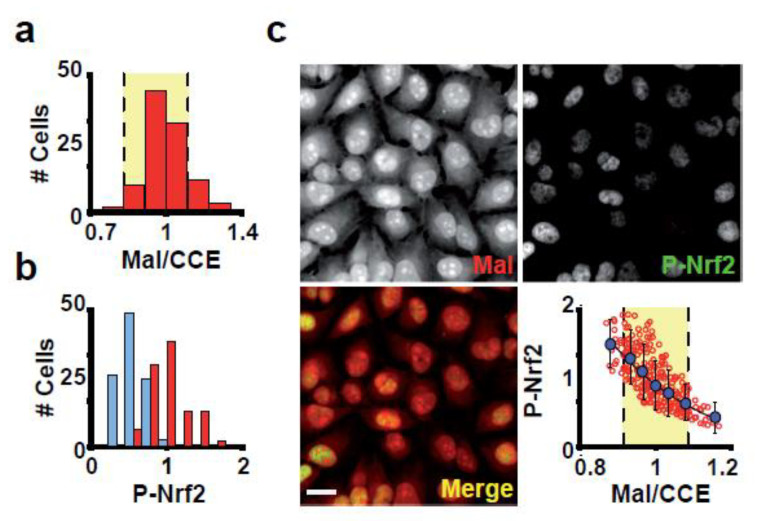
The signal of P-Nrf2 is related to the Mal-488/CCE-647 ratio. (**a**) The ratio of Mal-488/CCE-647 was different between cells. Only 10% of the cells displayed extreme Mal-488/CCE-647 values, falling outside of the yellow area representing cells with less than 10% variability. (**b**) P-Nrf2 intensity distribution in the nucleus of cells similar to panel c. Cells treated with diamide 50 μM for 1 h are represented by blue bars. (**c**) Cells stained with Mal-488, CCE-647, and P-Nrf2. The sub-panels show Mal-488 staining (Mal), P-Nrf2, a Merge panel of Mal-488 and P-Nrf2, and a scatter plot of P-Nrf2 versus the Mal-488/CCE-647 ratio. This plot shows that P-Nrf2 is strongly dependent on the amount of reduced Cys. Blue dots represent stratified data sampling, with samples taken every 0.05 unit of Mal-488/CCE-647 ratio. The stratified data can be fitted to an exponential decay with a ρ^2^ of 0.995. # Cells Stands for number of cells. Bar 10 μm.

**Figure 5 antioxidants-12-01274-f005:**
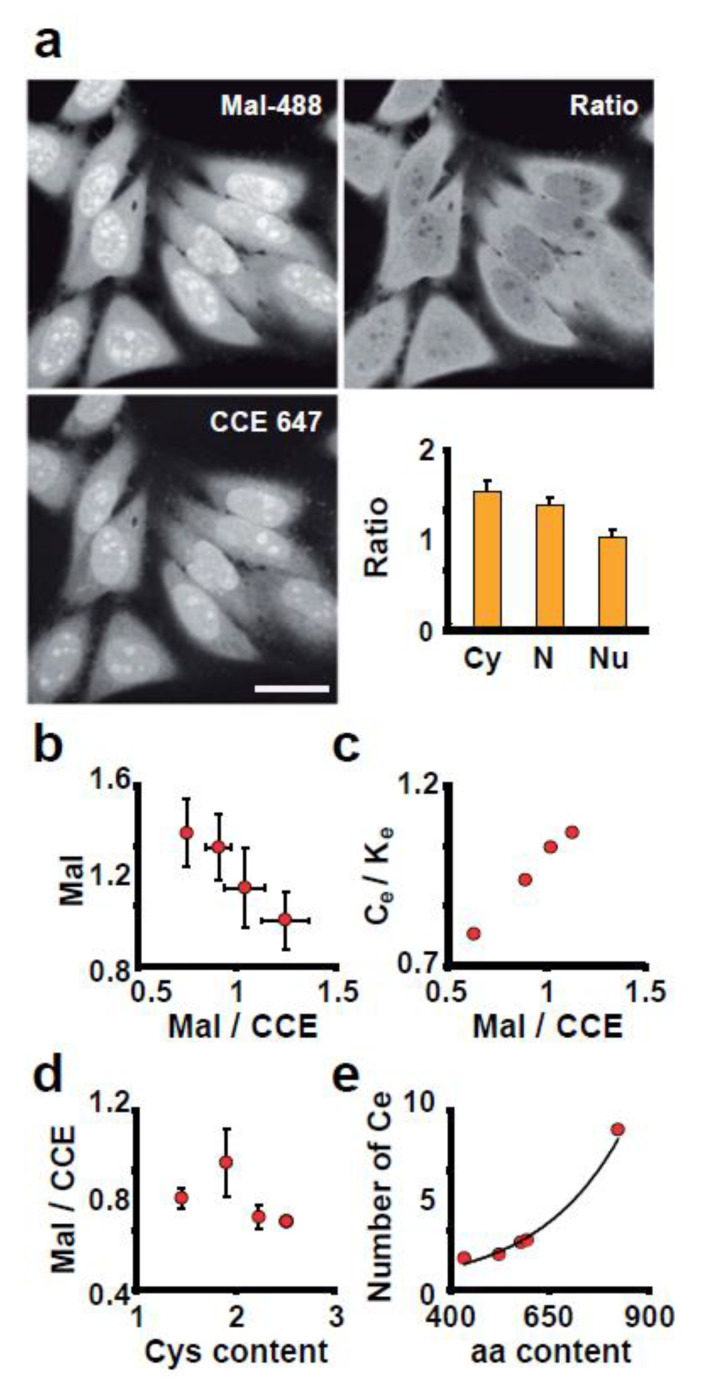
Each cell compartment has a distinct amount of reduced Cys. (**a**) Cells stained with Mal-488 and CCE-647. The ratio panel is created by dividing the Mal-488 signal by CCE-647 in each pixel. The bar chart shows the quantification of Mal-488/CCE-647 across different compartments (Cy is cytoplasm, N is nucleoplasm, and Nu is nucleolus). Bar 10 μm. (**b**) The relationship between Mal-488 content and the Mal-488/CCE-647 ratio in different compartments (nucleoplasm, nucleolus, cytoplasm, and mitochondria) is shown in this panel. The plot demonstrates an inverse relationship. (**c**) Plot of theoretical ratio of exposed Cys (Ce) and Lys (Ke); (Ce/Ke) versus Mal-488/CCE-647 values (the compartments were mitochondria, nucleoplasm, nucleolus, cytoplasm, and actin cytoskeleton). (**d**) A plot of the Mal-488/CCE-647 ratio versus the Cys content in different compartments. (**e**) Relationship between the average number of exposed Cys and the average number of aminoacids in proteins per compartment.

**Figure 6 antioxidants-12-01274-f006:**
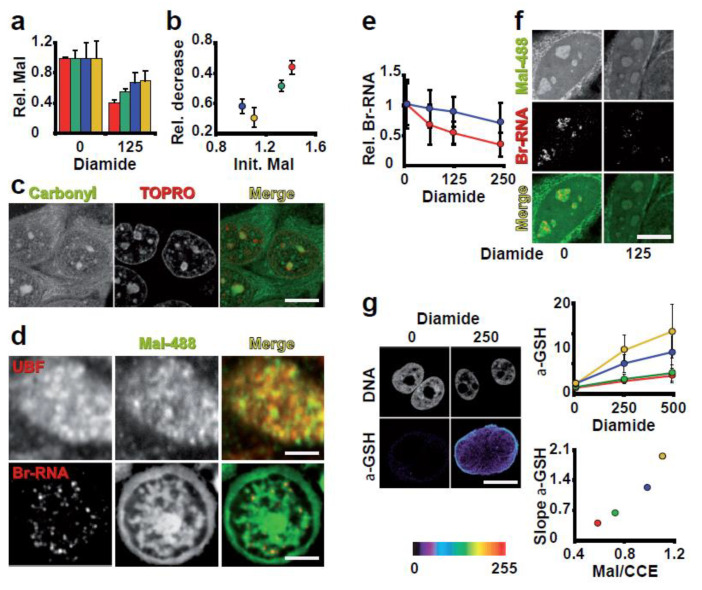
The amount of reduced Cys in a compartment determines its susceptibility to ROS. (**a**) This panel shows the quantification of Mal-488 staining/pixel in different compartments. The nucleolus is red, the nucleoplasm is green, the cytoplasm is blue, and the actin cytoskeleton is yellow. Each compartment’s standardisation was done separately. After 1 h of exposure to 125 μM diamide, all compartments showed a reduction in the staining signal, but it was not uniform. (**b**) Comparison between the initial level of Mal-488 per pixel in each compartment under investigation and the relative decline following diamide exposure. In this graph, the initial concentration of thiolates in a compartment is directly correlated with the relative signal loss. (**c**) Proteins stained to detect carbonyl groups. This panel demonstrates that the cytoplasm and nucleolus contain more carbonyls than the nucleoplasm does. Bar 5 μm. (**d**) Using Mal-488 to stain UBF or Br-RNA. This panel illustrates the relationship between high Mal-488 staining areas and transcription foci for both RNA pol I (UBF) and RNA pol II. Bars: for UBF 300 nm and for Br-RNA 2μm. (**e**) Evaluation of the Br-RNA synthesis activity’s sensitivity to a 1 h diamide exposure. Different effects are seen on the RNA pol II (blue) and RNA pol I (red) activities. (**f**) The RNA pol I activity (Br-RNA)’s sensitivity to exposure to diamide is demonstrated in this panel. Bar 5 μm. (**g**) Following a 1 h exposure to 250 μM diamide, the glutathionylated proteins are stained in the first panel. Only cells that have been exposed to light are clearly stained, and the GSH signal gathers near the cytoplasmic edge. Bar 10 μm. Quantitative analysis of GSH staining in various compartments is shown in the upper graph. Yellow represents the actin cytoskeleton, green the nucleolus, blue the cytoplasm, and red the nucleolus. The lower plot shows the relationship between the slope of the lines in the top panel and the Mal-488/CCE-647 ratio. This graph clearly shows a direct relationship between compartment glutathionylation and thiolate group concentration per protein. 2.5. Intra-nuclear reactive thiol compartmentalization.

**Figure 7 antioxidants-12-01274-f007:**
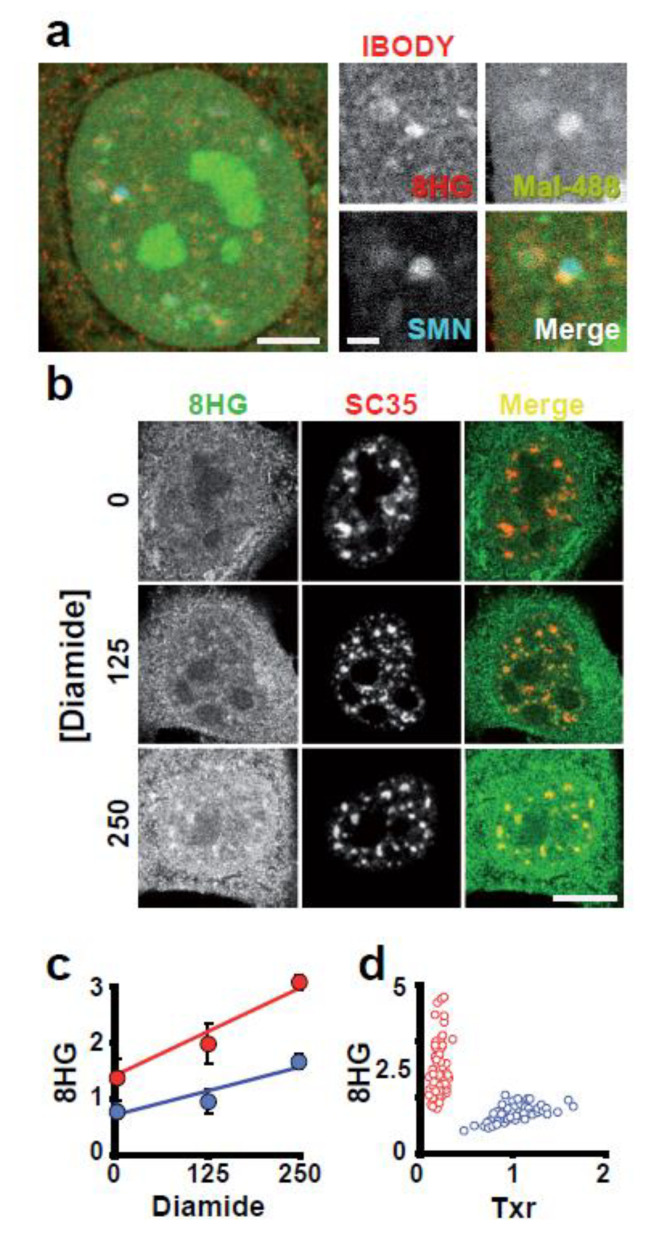
The cell nucleus compartmentalises oxidised RNA. (**a**) Mal-488, 8HG (oxidised RNA), and SMN bodies were used to stain the cell nucleus. A zoom of the new IBODY, which is close to SMN and is highly concentrated in oxidised RNA, is shown in the four panels insert. Bar 2 μm. (**b**) At the SC35 nuclear speckles, oxidised RNA accumulate. The different panels demonstrate the rise in oxidised RNA in the cell following a 1 h exposure to diamide. It is easy to see that the 8HG signal in the SC35 nuclear speckles has increased. Bar 5 μm. (**c**) Quantification of the 8HG signal/pixel in the nucleoplasm (blue) and nuclear speckles (red) of the SC35 image. The SC35 nuclear speckles line has a slope that is steeper than the nucleoplasms line, indicating that oxidised RNA has accumulated selectively in the speckles. (**d**) Measuring the 8HG signal and Trx in specific cells. According to the Section 2, red circles represent Trx knockdown cells and blue circles represent control cells.

**Table 1 antioxidants-12-01274-t001:** With details of the Cellular compartments studied.

Cellular Component	GO Accession	Size	aa Average per Protein
SC35 Nuclear Speckles	GO:0016607	129	720.74
Nucleolus	GO:0005730	520	581.21
Nucleus	GO:0005634	5523	624.19
Mitochondria	GO:0005739	1353	429.08
Cytoskeleton	GO:0005856	1593	830.95
Cytoplasm	GO:0005737	8375	595.86
Extracellular	GO:0005576	2227	521.05
Proteome	Non available	20,237	557.75

## Data Availability

Not applicable.

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
