# Peer review of "Each Cellular Compartment Has a Characteristic Protein Reactive Cysteine Ratio Determining Its Sensitivity to Oxidation"

_antioxidants, 2023, doi:10.3390/antiox12061274_

Round 1

Reviewer 1 Report

The manuscript entitled “Each cellular compartment has a characteristic protein reactive cysteine ratio determining its sensitivity to oxidation” by Ricardo Pires das Neves et al. shows that the proteins in each subcellular compartment contains a characteristic Cys amount. Using a fluorescent assay for SH in thiolate form and amino groups in proteins, the authors show that the thiolate content correlates with ROS sensitivity and signaling properties of each compartment. The highest absolute thiolate concentration was found in the nucleolus, followed by the nucleoplasm and cytoplasm whereas protein thiolate groups per protein showed an inverse pattern. 

 The first part of this paper is interesting and its aim i.e. the determination of the different amount of protein cysteines in each subcellular compartment, deserves publication on Antioxidant.  These information complete a previous paper (see ref. 7) which describes the total amount of reactive proteins in cells. Only a few concerns:

1.    The authors refer to the pKa of GSH or protein cysteines as the main determinant of their reactivity.  This is not correct. A recent paper discovers that pKa variation cannot influence the reactivity more that three-four times (New Factors Enhancing the Reactivity of Cysteines in Molten Globule-Like Structures. G, Cattani G, Bocedi A, Ricci G. Int J Mol Sci. 21, 6949 (2020)). Other factors can increase the reactivity hundred or thousand times.  

2.   The sentence “maleimide reacts with thiolate groups; cysteine in the GSH molecule have a pKa of 8.8 which means that at the pH of the reaction (7.0) less than 10% of thiol groups are deprotonated and then reactive” (lines 142-144) suggests that the authors do not have a clear knowledge of a chemical equilibrium.  Even if only 10% of GSH is in the GS- form, the reaction with maleimide can shift the equilibrium GSH ----GS-  very rapidly toward right….

3.    The sentence “even when exposed the reactivity can be hampered by a basic pKa for the thiolic group (which means not been protonated at physiological pH)(line 184)   is likely false: ….Which means not been DEPROTONATED at physiological..)

The second part of the manuscript, (Cap 2.4 and 2.5) is more questionable.

The reaction of thiols with diazenecarbonyl derivatives—such as diamide—occurs in two observable stages, with thiolate anions (RS−) as the reactive species. The reaction proceeds via addition and displacement steps. In the case of GSH, the GS− anion adds to the diazene double bond to form a sulfenylhydrazine, which, in a second step, reacts with a second GS− anion at sulfur to yield GSSG and a hydrazine. In the case of a protein cysteine, the pathway is similar to yield a protein disulfide or a mixed disulfide protein-SS-G.

The use of diamide is therefore a very specific reagent toward intracellular free and protein cysteines.  Diamide triggers ROS generation. But was the incubation with diamide performed in PBS or in a nutrient medium? This must be reported clearly in Materials and Methods.  If only PBS was used, than  the efficient redox buffer represented by GSH is de facto made inactive because, in the absence of nutrients, NADPH cannot be produced and  glutathione reductase cannot reduce GSSG. 

In conclusion, all correlations between the distribution of reactive thiols and ROS sensitivity (Chapter 2.4) may be altered in the absence of a GSH recycling mechanism.   We are sure that the GSH recycling mechanism opposes any deleterious oxidative event on protein cysteines (most of them are functional residues involved in catalytic mechanism). The eminent role of GSH as the true red-ox buffer is well known and underlined by the existence of three active  enzyme i.e. glutathione peroxidase, glutathione transferase with selenium independent activity toward organic hydroperoxides, and glutathione reductase.   Thus the sentence .” This, together with the fact that a comprehensive analysis of the total cellular thiol pools showed that reactive SH groups in proteins represent around 70% of the total reactive SH groups available in the cell, questions the direct redox buffer function for GSH.(lines396-398”), must be revised. Even the sentence between lines 481-489 in the Discussion is not shareable. The accumulation off GSH inside the nucleus is well documented even if the mechanism is unknown. In addition, if the GSH reducing system is active, it is not surprising that the glutathionylation is low in the nucleus even in the presence of high GSH.  In summary, I do not believe that the proposed method predict the range of susceptibility to the damage by ROS for each cell compartment excluding the analysis of the principal actor i.e.  the GSH/GSSG system (lines 505-508) 

The discussion must be rcorrected and shortened consistently.

Author Response

The manuscript entitled “Each cellular compartment has a characteristic protein reactive cysteine ratio determining its sensitivity to oxidation” by Ricardo Pires das Neves et al. shows that the proteins in each subcellular compartment contains a characteristic Cys amount. Using a fluorescent assay for SH in thiolate form and amino groups in proteins, the authors show that the thiolate content correlates with ROS sensitivity and signaling properties of each compartment. The highest absolute thiolate concentration was found in the nucleolus, followed by the nucleoplasm and cytoplasm whereas protein thiolate groups per protein showed an inverse pattern. 

 The first part of this paper is interesting and its aim i.e. the determination of the different amount of protein cysteines in each subcellular compartment, deserves publication on Antioxidant.  These information complete a previous paper (see ref. 7) which describes the total amount of reactive proteins in cells. Only a few concerns:

  1. The authors refer to the pKa of GSH or protein cysteines as the main determinant of their reactivity.  This is not correct. A recent paper discovers that pKa variation cannot influence the reactivity more that three-four times (New Factors Enhancing the Reactivity of Cysteines in Molten Globule-Like Structures. G, Cattani G, Bocedi A, Ricci G. Int J Mol Sci. 21, 6949 (2020)). Other factors can increase the reactivity hundred or thousand times.  

 This sentence has been corrected

  1. The sentence “maleimide reacts with thiolate groups; cysteine in the GSH molecule have a pKa of 8.8 which means that at the pH of the reaction (7.0) less than 10% of thiol groups are deprotonated and then reactive”(lines 142-144) suggests that the authors do not have a clear knowledge of a chemical equilibrium.  Even if only 10% of GSH is in the GS- form, the reaction with maleimide can shift the equilibrium GSH ----GS-  very rapidly toward right….

 This sentence has been eliminated from the manuscript

  1.  The sentence “even when exposed the reactivity can be hampered by a basic pKa for the thiolic group (which means not been protonated at physiological pH)(line 184)   is likely false: ….Which means not been DEPROTONATED at physiological..)

This sentence has been eliminated from the manuscript

The second part of the manuscript, (Cap 2.4 and 2.5) is more questionable.

The reaction of thiols with diazenecarbonyl derivatives—such as diamide—occurs in two observable stages, with thiolate anions (RS−) as the reactive species. The reaction proceeds via addition and displacement steps. In the case of GSH, the GS− anion adds to the diazene double bond to form a sulfenylhydrazine, which, in a second step, reacts with a second GS− anion at sulfur to yield GSSG and a hydrazine. In the case of a protein cysteine, the pathway is similar to yield a protein disulfide or a mixed disulfide protein-SS-G.

The use of diamide is therefore a very specific reagent toward intracellular free and protein cysteines.  Diamide triggers ROS generation. But was the incubation with diamide performed in PBS or in a nutrient medium?

It was done in DMEM. This was specified in Material and Methods.

This must be reported clearly in Materials and Methods.  If only PBS was used, than  the efficient redox buffer represented by GSH is de facto made inactive because, in the absence of nutrients, NADPH cannot be produced and  glutathione reductase cannot reduce GSSG. 

In conclusion, all correlations between the distribution of reactive thiols and ROS sensitivity (Chapter 2.4) may be altered in the absence of a GSH recycling mechanism.   We are sure that the GSH recycling mechanism opposes any deleterious oxidative event on protein cysteines (most of them are functional residues involved in catalytic mechanism). The eminent role of GSH as the true red-ox buffer is well known and underlined by the existence of three active  enzyme i.e. glutathione peroxidase, glutathione transferase with selenium independent activity toward organic hydroperoxides, and glutathione reductase.   Thus the sentence .” This, together with the fact that a comprehensive analysis of the total cellular thiol pools showed that reactive SH groups in proteins represent around 70% of the total reactive SH groups available in the cell, questions the direct redox buffer function for GSH.(lines396-398”), must be revised.

We have changed the sentence

Even the sentence between lines 481-489 in the Discussion is not shareable. The accumulation off GSH inside the nucleus is well documented even if the mechanism is unknown.

I disagree with the referee, there is no direct evidence for the existence of GSH in the cell nucleus. All papers proposing the existence of a nuclear GSH pool are based on the use of reagents that react with -SH groups (Mercury orange, monochloro bimane, and 5-chloromethylfluorescein diacetate (CMFDA)).  Many of them have resulted in conflicting conclusions.  The most recent evidences for GSH accumulation in the cell nucleus come from the use of CMFDA, a cell-permeable compound that is retained inside the cell after being transformed by cellular esterases, presumably due to its ability to react with -SH groups. As a result, when we analyse the fluorescence due to CMFDA, there are many uncertainties that cast doubt on its usefulness. Intracellular esterase activity, for example, changes with proliferation and/or cell cycle status, which affects the intracellular signal of CMFDA.

In addition, if the GSH reducing system is active, it is not surprising that the glutathionylation is low in the nucleus even in the presence of high GSH.  

In summary, I do not believe that the proposed method predict the range of susceptibility to the damage by ROS for each cell compartment excluding the analysis of the principal actor i.e.  the GSH/GSSG system (lines 505-508).

Sorry to disagree with the referee, but I don't think GSH/GSSG is the main actor, it is one more, very important but not the most important. Paraphrasing Ursula Jakob “GSH works primarily by providing the reducing equivalents for antioxidant enzymes, including glutathione peroxidases” (doi.org/10.1016/j.freeradbiomed.2019.05.035)

Reviewer 2 Report

Review of the paper entitled „Each cellular compartment has a characteristic protein reactive cysteine ratio determining its sensitivity to oxidation by Ricardo Pires das Neves, Mónica Chagoyen, Antonio Martinez- Lorente, Carlos Iñiguez, Juana Calabuig and Francisco J. Iborra.

     The human genome encodes about 214,000 cysteine (Cys). Cys is widely distributed among proteins, with most expressing at least one and many having multiple coordinated Cys and Cys-rich domains. Cys-derived protein thiol groups can be reduced or oxidized as disulfides and mixed disulfides when conjugated with low molecular weight thiols: glutathione, cysteine, homocysteine and γ-glutamylcysteine.

     The authors pay attention that Cys is an important amino acid which plays a major role in the protein structure and function by changing the redox status. This research showed that the proteins in each subcellular compartment contain a characteristic amount of Cys residues.

     The topic is attractive, the results obtained by the authors are very interesting, however, the manuscript needs a significant  improvement.

My comments

Materials and Methods. The authors should describe the used methods in a more transparent manner. First of all the principle of the methods should be stated.

For examples

·         The pH-dependent ionization of the nucleophilic cysteine thiol was followed by the specific absorbance of the thiolate anion at 240 nm. As a reference, the pH-dependent absorbance for the oxidized form of the protein was monitored [Heras et al. J Biol Chem. 2008 Feb 15;283(7):4261-71].

·         The 4-acetamido-4'-maleimidylstilbene-2,2'- disulfonic acid (AMS) is a thiol-reactive reagent that is water soluble, with high polarity and membrane impermeability. The polyethyleneglycol (PEG)-conjugated-malemide (MalPEG5000) to alkylate free thiol of protein can also be used instead of AMS. In both treatments, samples were analyzed by SDS-PAGE and immunoblotting with appropriate antiprotein serum. AMS increases protein mass by 490 Da and MalPEG by 5,000 Da, observed as band shifts [Depuydt et al. Science. 2009 Nov 20;326(5956):1109-11].

      I also suggest include an appropriate scheme to show the method of protein reactive cysteine ratio analysis.

Results and Discussion

     In my opinion in this section too many elements of description and discussion is included. Results section may include some elements of discussion of the obtained results, however if separate discussion section exists such elements should be limited. For example, lines 142-153 – this is discussion with cited other references, not the obtained results; lines 111-118 – there are not the results. The next examples: lines 181-188, 193-196 or 208-213, 320-330. Too much information in this section makes difficult to follow the presented results.

Other comments

-          In some cases the English spelling is incorrect, i.e. “some times” instead “sometimes”; “aminoacid” instead “amino acid”

-          In the Antioxidants journal the references should be cited in the manuscript  in square brackets not as superscripts

-          There are some grammar mistakes, i.e. line 42: …species, which act (not acts) as toxicants…

-          Line 37: Since the abbreviation ROS was introduced in the line 32 it should then be used consistently in the manuscript

-          Line 46: neutrophilic or rather nucleophilic?

-          In my opinion the term “thiol group” would be better than “thiolic group”

-          Line 54: “Fort he exposed reasond…” – please correct

-          Line 55: glutathione instead Glutathione

-          Line 57: “ROS-induced protein inactivation” instead “ROS protein inactivation”

-          Lines 85-91: This information was included in the introduction, there is no need to repeat it in the results

-          The abbreviations Cys and Lys once introduced should be consistently used in the manuscript (see line 116, 143, 202 and other)

-          Before SH group, hyphen should be used (–SH)

-          Lines 166-167 – This information was given earlier, there is no need to repeat it

-          Figure 2 legend – there is too much information, which was described in results section. The value of p can be given as p<0,0001)

-          What is P-Nrf2? Is it phosphorylated Nrf2? – it should be clearly stated

-          Line 227-228 – the last part of this sentence is grammatically incorrect 

-          Line 272: The abbreviations Ce and Ke are not explained, moreover in the line 518 Cexp and Kexp were used

-          Line 276: “The distribution of reactive thiol groups correlates with sensitivity to ROS” rather than “correlates with ROS sensitivity”

-          Line 294: “ interpretation of that is…” – as mentioned earlier, there is rather discussion than results

-          Line 300 and 305: explanation of UBF and BrUTP are needed

-          Line 325 – the glutathionylation of proteins has been studied, however in Materials and Methods I have not found description of this method

-          Line 355, 358 – the abbreviations SMN, PML, PTF, OCT1 are not explained

-          Line 382 – there is a double dot at the end of the sentence

-          Line 463  – the abbreviation hnRNP is not explained

-          Line 480 –  kDa instead KD

-          Line 518 – the sentence is interrupted

-          Line 543 – the abbreviation TRAIL should be explained in the text

-          Line 463  – the abbreviation DRB is not explained

-          Line 561  – the abbreviation 8HG is not explained

-          Line 568 – the abbreviation Trx should be used

-          Line 579  – the abbreviation DAPI is not explained

-          Line 603 – the abbreviation for thioredoxin is Trx rather than Txr

-          Lines 589-594 – Acknowledgments: Why the number of grant number is given twice?

-          There should be no dots at the end of titles, please, correct it.

Author Response

Review of the paper entitled „Each cellular compartment has a characteristic protein reactive cysteine ratio determining its sensitivity to oxidation by Ricardo Pires das Neves, Mónica Chagoyen, Antonio Martinez- Lorente, Carlos Iñiguez, Juana Calabuig and Francisco J. Iborra.

     The human genome encodes about 214,000 cysteine (Cys). Cys is widely distributed among proteins, with most expressing at least one and many having multiple coordinated Cys and Cys-rich domains. Cys-derived protein thiol groups can be reduced or oxidized as disulfides and mixed disulfides when conjugated with low molecular weight thiols: glutathione, cysteine, homocysteine and γ-glutamylcysteine.

     The authors pay attention that Cys is an important amino acid which plays a major role in the protein structure and function by changing the redox status. This research showed that the proteins in each subcellular compartment contain a characteristic amount of Cys residues.

     The topic is attractive, the results obtained by the authors are very interesting, however, the manuscript needs a significant  improvement.

My comments

Materials and Methods. The authors should describe the used methods in a more transparent manner. First of all the principle of the methods should be stated.

For examples

  • The pH-dependent ionization of the nucleophilic cysteine thiol was followed by the specific absorbance of the thiolate anion at 240 nm. As a reference, the pH-dependent absorbance for the oxidized form of the protein was monitored [Heras et al. J Biol Chem. 2008 Feb 15;283(7):4261-71].
  • The 4-acetamido-4'-maleimidylstilbene-2,2'- disulfonic acid (AMS) is a thiol-reactive reagent that is water soluble, with high polarity and membrane impermeability. The polyethyleneglycol (PEG)-conjugated-malemide (MalPEG5000) to alkylate free thiol of protein can also be used instead of AMS. In both treatments, samples were analyzed by SDS-PAGE and immunoblotting with appropriate antiprotein serum. AMS increases protein mass by 490 Da and MalPEG by 5,000 Da, observed as band shifts [Depuydt et al. Science. 2009 Nov 20;326(5956):1109-11].

      I also suggest include an appropriate scheme to show the method of protein reactive cysteine ratio analysis.

Results and Discussion

     In my opinion in this section too many elements of description and discussion is included. Results section may include some elements of discussion of the obtained results, however if separate discussion section exists such elements should be limited. For example, lines 142-153 – this is discussion with cited other references, not the obtained results; lines 111-118 – there are not the results. The next examples: lines 181-188, 193-196 or 208-213, 320-330. Too much information in this section makes difficult to follow the presented results.

Other comments

-          In some cases the English spelling is incorrect, i.e. “some times” instead “sometimes”; “aminoacid” instead “amino acid”

-          In the Antioxidants journal the references should be cited in the manuscript  in square brackets not as superscripts

-          There are some grammar mistakes, i.e. line 42: …species, which act (not acts) as toxicants…

-          Line 37: Since the abbreviation ROS was introduced in the line 32 it should then be used consistently in the manuscript

-          Line 46: neutrophilic or rather nucleophilic?

-          In my opinion the term “thiol group” would be better than “thiolic group”

-          Line 54: “Fort he exposed reasond…” – please correct

-          Line 55: glutathione instead Glutathione

-          Line 57: “ROS-induced protein inactivation” instead “ROS protein inactivation”

-          Lines 85-91: This information was included in the introduction, there is no need to repeat it in the results

-          The abbreviations Cys and Lys once introduced should be consistently used in the manuscript (see line 116, 143, 202 and other)

-          Before SH group, hyphen should be used (–SH)

-          Lines 166-167 – This information was given earlier, there is no need to repeat it

-          Figure 2 legend – there is too much information, which was described in results section. The value of p can be given as p<0,0001)

-          What is P-Nrf2? Is it phosphorylated Nrf2? – it should be clearly stated

-          Line 227-228 – the last part of this sentence is grammatically incorrect 

-          Line 272: The abbreviations Ce and Ke are not explained, moreover in the line 518 Cexp and Kexp were used

-          Line 276: “The distribution of reactive thiol groups correlates with sensitivity to ROS” rather than “correlates with ROS sensitivity”

-          Line 294: “ interpretation of that is…” – as mentioned earlier, there is rather discussion than results

-          Line 300 and 305: explanation of UBF and BrUTP are needed

-          Line 325 – the glutathionylation of proteins has been studied, however in Materials and Methods I have not found description of this method

The process of glutathionylation is assessed by the immunostaining with GSH antibodies. Because GSH can not be fixed unless that is bound to proteins (glutathionylation)

-          Line 355, 358 – the abbreviations SMN, PML, PTF, OCT1 are not explained

-          Line 382 – there is a double dot at the end of the sentence

-          Line 463  – the abbreviation hnRNP is not explained

-          Line 480 –  kDa instead KD

-          Line 518 – the sentence is interrupted

-          Line 543 – the abbreviation TRAIL should be explained in the text

-          Line 463  – the abbreviation DRB is not explained

-          Line 561  – the abbreviation 8HG is not explained

-          Line 568 – the abbreviation Trx should be used

-          Line 579  – the abbreviation DAPI is not explained

-          Line 603 – the abbreviation for thioredoxin is Trx rather than Txr

-          Lines 589-594 – Acknowledgments: Why the number of grant number is given twice?

-          There should be no dots at the end of titles, please, correct it.

All these corrections have been done

Reviewer 3 Report

In this MS the authors are attempting to determine the relative sensitivity of Protein-Cys thiols, in various cellular compartments, cytosol, nucleus, nucleolus, to oxidative stress.  The authors claim that the novel aspect of this study is that they are determining relative Cys reactivity from imaging of HeLa cells that are labelled with a fluorescent thiol probe Maleimide-488.  The authors have done many controls to determine that what they are imaging are free protein Cys thiols. The amount of data presented is voluminous and well presented.  However there is a big flaw in experimental design.  They are labeling the thiols after fixing with paraformaldehyde.  The fixation method is 4% parformaldehyde for 10-15 minutes.  This is a lot of paraformaldehyde! Paraformalehyde alkylates thiols. the title of this study should be "The monitoring of paraformaldehyde reactivity with protein thiols using fluorescent thiol probes and imaging."  The information the authors are after can be obtained with permeabilized non fixed cells.   Another concern is the use of HeLa cells.  Why not use non cancerous cells? 

Author Response

In this MS the authors are attempting to determine the relative sensitivity of Protein-Cys thiols, in various cellular compartments, cytosol, nucleus, nucleolus, to oxidative stress.  The authors claim that the novel aspect of this study is that they are determining relative Cys reactivity from imaging of HeLa cells that are labelled with a fluorescent thiol probe Maleimide-488.  The authors have done many controls to determine that what they are imaging are free protein Cys thiols. The amount of data presented is voluminous and well presented.  However there is a big flaw in experimental design.  They are labeling the thiols after fixing with paraformaldehyde.  The fixation method is 4% parformaldehyde for 10-15 minutes.  This is a lot of paraformaldehyde! Paraformalehyde alkylates thiols. the title of this study should be "The monitoring of paraformaldehyde reactivity with protein thiols using fluorescent thiol probes and imaging."  The information the authors are after can be obtained with permeabilized non fixed cells.   Another concern is the use of HeLa cells.  Why not use non cancerous cells? 

The high reactivity of formaldehyde with the thiol group of Cys is well known, but the chemistry and kinetics are complicated. Some of the reactive Cys are likely to have reacted with formaldehyde during the fixation process. However, the fact that we can detect maleimide reactivity with cellular proteins and that all of the controls we performed indicate that this technique detects reduced Cys. This technique is not ideal for quantifying the absolute amount of reduced Cys in proteins, but it detects a fraction of them that correlates well with the expected concentration in each compartment (figure 5). We have incubated permeabilized un fixed cell with fluorescent maleimide and the results were the same that the ones obtained in fixed cells.

While the reaction conditions are not ideal for detecting all of the reduced Cys in the cell, they are sufficient for informing us about the subcellular heterogeneity of reduced protein distribution.

Round 2

Reviewer 1 Report

The revised paper can be accepted in the present form.

Author Response

ok

Reviewer 2 Report

The Authors have corrected some mistakes (i.e. references, acknowledgments, some abbreviations) but they did not address other suggestions. Generally, I am disappointed with the revised version of this paper. The authors did not mark the changes in the text, and in their reply to the reviewer they did not refer to individual comments. The general sentence in their reply "All these corrections have been done", is not true. There is still lack of the description of protein glutathionylation in Materials and Methods. The suggestion to add the scheme presenting the method has been ignored by Authors without explanation. The abbreviations have been added to the Abbreviations list but I think it will be nice to add them in the text when are used firs time. Cysteine and lysine have not been changed by Cys and Lys. Interestingly, during the revision subheadings 2.2 and 2.5 disappeared...

Overall, I don't think the improvement is enough.

Author Response

The Authors have corrected some mistakes (i.e. references, acknowledgments, some abbreviations) but they did not address other suggestions. Generally, I am disappointed with the revised version of this paper. The authors did not mark the changes in the text, and in their reply to the reviewer they did not refer to individual comments.

We regret not having marked the changes. in this new submission we provide a version of the manuscript with the marked changes, for easy follow-up.

The general sentence in their reply "All these corrections have been done", is not true.

There is still lack of the description of protein glutathionylation in Materials and Methods.

We have introduced the required description.

The suggestion to add the scheme presenting the method has been ignored by Authors without explanation.

We have ignored this requirement because we believe it is not necessary when using a technique that has already been described in another publication (ref 19).

The abbreviations have been added to the Abbreviations list but I think it will be nice to add them in the text when are used firs time. Cysteine and lysine have not been changed by Cys and Lys. Interestingly, during the revision subheadings 2.2 and 2.5 disappeared...

Reviewer 3 Report

The authors' comments do not change the decision to reject.

Author Response

We agree that formaldehyde reacts with the thiol groups of proteins. So our binding protocol should alter the detection of SH groups in proteins. However, as the chemistry is complex, it is very difficult to know exactly the magnitude of such a modification. Therefore, even if the conditions were not optimal, we decided to continue with the staining, so we carried out all the experiments described in figure 2.
In addition, we tested different types of fixation, such as the use of methanol, which we discarded for other reasons such as poor protein retention. As referenced in the materials and methods section, this histochemical staining was described in an article published in 2011 (DOI: 10.1089/ars.2010.3629). We have only further validated this technique and are convinced that it is the reduced SH groups in the proteins that stain. 
Following your suggestion, we carried out staining on saponin permeabilised cells, without fixation, and observed that the staining pattern is comparable to that obtained with formaldehyde-fixed cells (intense labelling in the nucleolus, followed by the nucleoplasm and less staining in the cytoplasm). This control has been introduced in panel h of figure 2.